

# Weather and nest cavity characteristics influence fecundity in mountain chickadees

Andrea R. Norris[1,2], Kathy Martin[1,2] and Kristina L. Cockle[2,3]

[1] Science and Technology Branch, Wildlife Research Division, Environment and Climate Change Canada, Delta, BC, Canada

[2] Department of Forest and Conservation Sciences, Faculty of Forestry, University of British Columbia, Vancouver, BC, Canada

[3] CONICET-Universidad Nacional de Misiones, Instituto de Biología Subtropical, Puerto Iguazú, Misiones, Argentina

Corresponding author
Andrea R. Norris,
andrea.norris@ec.gc.ca

## ABSTRACT

**Background**. Examining direct and indirect effects on reproduction at multiple scales allows for a broad understanding of species' resilience to environmental change. We examine how the fecundity of the mountain chickadee (*Poecile gambeli*), a secondary cavity-nesting, insectivorous bird, varied in relation to factors at three scales: regional weather conditions, regional- and site-level food availability, site-level community dynamics, and nest-level cavity characteristics. We hypothesized that earlier laying dates and higher fecundity (clutch size, nest survival, brood size) would be associated with milder climatic conditions, increased food from insect outbreaks, lower densities of conspecifics and nest predators (red squirrel; *Tamiasciurus hudsonicus*), and safer (smaller, higher) cavities.

**Methods**. We collected data on laying date, clutch size, brood size, nest fate (success/failure), and cavity characteristics from 513 mountain chickadee nests in tree cavities in temperate mixed coniferous-broadleaf forest in interior British Columbia, Canada, from 2000 to 2011. We surveyed annual abundances of mountain chickadees and squirrels using repeated point counts, and mountain pine beetle (*Dendroctonus ponderosae*) and lepidopteran defoliators by monitoring host trees and by using regional-scale aerial overview forest insect survey data. We used weather data (temperature, rain, snow) from a local Environment and Climate Change Canada weather station. We modeled laying date, clutch size, daily nest survival, and brood size as a function of predictors at regional-, site-, and nest-scales.

**Results and Conclusions**. Measures of fecundity varied dramatically across years and spatial scales. At the regional (study-wide) scale, chickadees laid earlier and larger first clutches in warmer springs with minimal storms, and daily nest survival (DSR) increased with a 2-year lag in growing season temperature. Despite a doubling of mountain chickadee density that roughly accompanied the outbreaks of mountain pine beetle and lepidopteran defoliators, we found little evidence at the site scale that fecundity was influenced by insect availability, conspecific density, or predator density. At the nest scale, DSR and brood size increased with clutch size but DSR declined with nest cavity size indicating a positive reproductive effect of small-bodied cavity excavators. Double-brooding, rare in chickadees, occurred frequently in 2005 and 2007, coinciding with early breeding, high food availability from insect outbreaks, and warm spring temperatures with 0-1 spring storms. Our results support the idea that fecundity

in secondary cavity-nesting species is impacted directly and indirectly by weather, and indirectly through changes in community dynamics (*via* cavity resource supply). We stress the importance of adopting holistic, community-level study frameworks to refine our understanding of fecundity in opportunistic and climate-sensitive species in future.

# INTRODUCTION

Annual fecundity in birds (the number of young fledged per female per breeding season) comprises several components, including clutch size, proportion of eggs that hatch, nest success/failure, brood size (number of fledglings per nesting attempt), and number of nesting attempts per season (*Etterson et al., 2011*). These components of fecundity are often influenced by weather conditions early in the breeding season, *via* impacts on nest phenology (*Drake & Martin, 2018*; *Kozlovsky et al., 2018*; *de Zwaan et al., 2019*; *de Zwaan et al., 2022*; *Martin et al., 2020*). Weather can also affect phenology and fecundity indirectly, by altering community dynamics (*e.g.*, competitor and predator populations), food availability, and body condition of adults (*Descamps et al., 2011*; *Norris & Martin, 2014*; *de Zwaan et al., 2019*; *de Zwaan et al., 2020*). Studies that examine how climate impacts fecundity directly and indirectly through local habitat conditions and community-level processes can offer a more comprehensive understanding of climate-sensitive species.

Resident, secondary cavity nesting birds (those that nest in pre-existing tree cavities), in northern temperate forests, represent a good study system for examining fecundity variation, because of their large clutches and reproductive plasticity. Resident and short-distant migratory species can match their breeding decisions to local environmental conditions prior to the breeding season (*Lack, 1954*; *Alerstam, Hedenström & Åkesson, 2003*; *Salewski & Bruderer, 2007*; *Shaw & Couzin, 2013*; *Drake & Martin, 2018*). Relative to open-cup nesters, tree cavity nesters can produce more offspring per nesting attempt, likely because cavities protect offspring from nest predators (*Martin, 1995*; *Martin, 2014*). Furthermore, individuals that nest in pre-existing cavities can allocate time and energy saved on nest construction to laying earlier and larger clutches, or to adult body condition (*Martin, 1993*; *Mönkkönen & Martin, 2000*; *Wiebe, Koenig & Martin, 2007*; *Norris & Martin, 2014*). Most cavity-excavating species (*i.e.*, primarily woodpeckers and nuthatches) are also forest insectivores, and their population-level response to forest insect outbreaks can generate pulses of nest cavity production and availability, that lead to increases in secondary cavity-nesting species in subsequent years (legacy effect; *Cockle & Martin, 2015*; *Trzcinski et al., 2021*). During forest insect outbreaks of mountain pine beetle (Coleoptera: *Dendroctonus ponderosae*) and western spruce budworm (Lepidoptera: *Choristoneura occidentalis*) in western North America, the secondary cavity nesting forest insectivore, mountain chickadee (*Poecile gambeli*), tracked increases in populations of the small-bodied excavator, red-breasted nuthatch (*Sitta canadensis*), and shifted their nest-site

use from cavities with large entrances (mostly excavated by woodpeckers) to cavities with smaller entrances (mostly excavated by smaller woodpeckers and nuthatches) that were inaccessible to larger predators (*Norris, Drever & Martin, 2013*; *Cockle & Martin, 2015*). Mountain chickadee populations doubled in response to climate-induced pulses of nest cavities and food, and the shift in nest-site use signalled a positive legacy effect of the cavity pulse on chickadees (*Norris, Drever & Martin, 2013*), but the impacts of these cumulative changes in local environmental conditions on fecundity were not evaluated.

Warm spring weather and abundant food are expected to promote earlier breeding and higher fecundity in mountain chickadees (*Drake & Martin, 2018*; *Kozlovsky et al., 2018*; *Martin et al., 2020*) (Fig. 1A, Table 1). A seasonal decline in clutch size is common in temperate birds (*Slagsvold, 1982*; *Hochachka, 1990*), and clutch size is expected to decline with delays in clutch initiation, both within and among years. Laying date is expected to advance, and clutch size, nest survival, and brood size are expected to increase, with climate-driven pulses of forest insect prey. Food pulses can enable adults to lay a larger clutch, reduce starvation of individual nestlings, and decrease the time that adults need to spend foraging, which can increase the time available for nest defence, and reduce the risk of nest predation (*Rastogi, Zanette & Clinchy, 2006*). Warmer springs (Apr–Aug) and increased annual rainfall (Sept–Aug) can promote conifer seed masting events in subsequent years (*Rother, Veblen & Furman, 2015*; *Petrie et al., 2016*), and such events are predicted to allow chickadees to advance their laying date, and thus increase clutch and brood sizes. However, storm events (precipitation of >10 mm over 24 h) and deep snow in late winter/early spring may reduce the breeding condition of adults, contributing to delayed laying and smaller clutches. Spring storms during the breeding season may decrease the amount of time that adults spend foraging and feeding nestlings, leading to reductions in nest survival and brood size (*Kozlovsky et al., 2018*; *de Zwaan et al., 2019*).

Large numbers of conspecific competitors and nest predators are expected to negatively impact fecundity. Crowding and saturation of high quality habitats can reduce fecundity as conspecific densities increase (negative density dependence; (*Dhondt, Kempenaers & Adriaensen, 1992*; *Newton, 1998*; *Both & Visser, 2003*; *Rodenhouse et al., 2003*). Nest predation can impact fecundity directly by causing complete nest failure or by one or more predation events that reduce brood size but do not result in complete nest loss (partial nest depredation; *Ricklefs, 1969*; *Martin, 1993*) and indirectly by inducing adults to reduce investment when risk is high (top-down control; (*Skutch, 1949*; *Slagsvold, 1982*; *Fontaine & Martin, 2006*; *Travers et al., 2010*; *Ghalambor, Peluc & Martin, 2013*)). If risk of nest predation is an important driver of fecundity, mountain chickadees might lay later, smaller clutches, and fledge fewer young with increasing densities of one of their principal nest predators, red squirrels (*Tamiasciurus hudsonicus*; *Ghalambor & Martin, 2000*; *Martin, Aitken & Wiebe, 2004*; *Zanette et al., 2011*). They might also lay smaller clutches and fledge fewer young in larger cavities located closer to the ground (and therefore more readily accessible by larger-bodied predators; *Li & Martin, 1991*). However, if cavity size (space) limits brood size, mountain chickadees might lay larger clutches and broods in larger cavities (*Norris et al., 2018*). Nest survival might be positively correlated with clutch size because larger clutches allow nests to withstand multiple partial predation events (*Olmos,*
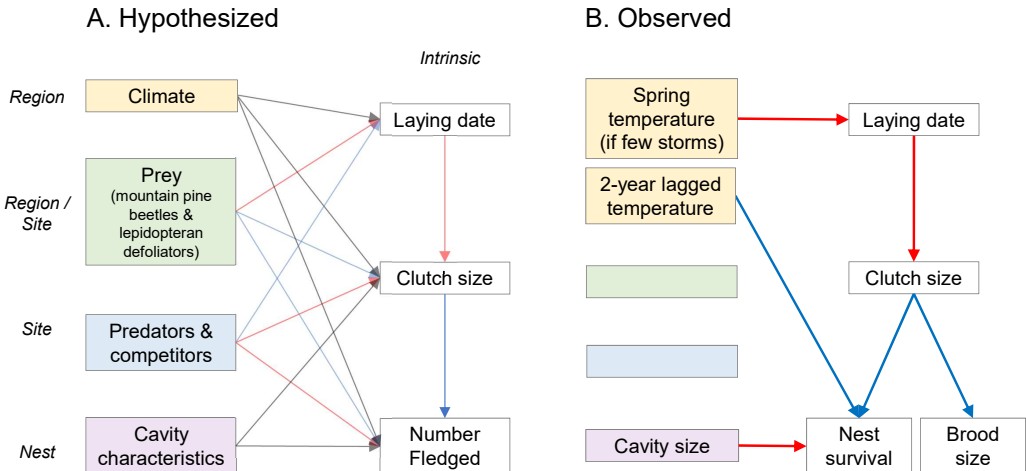

**Figure 1** **Hypothesized (A) and observed (B) drivers of fecundity of secondary cavity-nesting mountain chickadee (*Poecile gambeli*), in mixed broadleaf-coniferous forests at regional, site, and nest scales (italicized).** Left-hand columns represent external factors expected to influence fecundity. These external factors include, at regional scale, climate (winter snow depth, spring temperature, storms between January and April, 2-year-lagged temperature, 2-year-lagged rain) and winter prey abundance (mountain pine beetle larvae); at the site-scale, breeding season prey abundance (lepidopteran defoliators and mountain pine beetle), abundance of predators (red squirrel) and competitors (conspecifics); and at the nest-scale, cavity characteristics (height and size). Right-hand columns represent factors intrinsic to each nest (number fledged includes both nest survival and brood size of successful nests). Arrows point from potential drivers to measures of fecundity and indicate positive (blue), negative (red), or varied (grey) effects. (B) Solid red and blue arrows indicate relationships supported by model selection and parameter estimates whose 95% confidence intervals did not overlap zero (Tables 2 and 3).

*2003*), or because both nest survival and clutch size are correlated with an unmeasured external factor such as parental condition (*Slagsvold & Lifjeld, 1988*; *Slagsvold & Lifjeld, 1988*) or nest predation risk (*Slagsvold, 1982*; Table 1).

Our objective was to examine the direct and indirect effects of weather and other drivers for a species that opportunistically responds to environmental variation at multiple spatial and temporal scales, under natural conditions (*i.e.*, across a range of years and in natural tree cavities). We applied a broad, observational approach, combining data from multiple sources to test hypotheses at multiple spatial and temporal scales. We examined variation in timing (laying date) and fecundity (clutch size, daily nest survival, and brood size) of mountain chickadees nesting in tree cavities across a 12 year period (2000 to 2011) in interior British Columbia, encompassing outbreaks of mountain pine beetle (2002–2005) and defoliating lepidopterans (including western spruce budworm, 2002–2009). We then tested hypotheses about factors expected to influence components of mountain chickadee fecundity at the regional, site, and nest scales (Table 1, Fig. 1A).

## MATERIALS & METHODS

Field data were collected from May through July 2000–2011, at 23 sites within the warm and dry Interior Douglas-fir biogeoclimatic zone of central British Columbia, Canada.

**Table 1  Scale, hypotheses and predicted effects (±) of climate, food supply, predation risk, conspecific densities, and cavity characteristics on components of fecundity in mountain chickadees.** Components of fecundity are date of first egg laid (DFE), number of eggs laid (CS), daily nest survival (NS), and brood size (BS). Predictions are based on (*Drake & Martin, 2018*) (a); *Kozlovsky et al., 2018* (b); *Rother, Veblen & Furman, 2015* (c), *Petrie et al., 2016* (d); *Norris & Martin, 2014* (e); *Martin et al., 2020* (f); *Olmos, 2003* (g); *Slagsvold & Lifjeld, 1988*; (h); *Blomqvist, Johansson & Götmark, 1997* (i); *Slagsvold, 1982* (j).

| Scale of factors influencing fecundity | Hypotheses | Response ∼Predictor variables |
|---|---|---|
| Broad-scale, regional-level processes (via winter survival and breeding condition) | Mild winters and warm springs contribute to higher fecundity through improved winter survival[*] and body condition[*] and earlier breeding, but spring storms delay breeding, reducing fecundity[a,b] | DFE ∼ Spring temperature (−), Snow depth (+), Storms (+), Storms × Spring temperature CS/NS/BS ∼ Spring temperature (+), Snow depth (−), Storms (−), Storms × Spring temperature |
| | Drought and mild winters lead to conifer seed masting[*] and mountain pine beetle (MPB) outbreak, increasing winter food availability, leading to better condition[*], earlier breeding, and higher fecundity[c,d] | DFE ∼ Temperature (2 year lag, -), Rainfall (2 year lag, +), Pine beetle availability (−) CS/NS/BS ∼ Temp (2 year lag, +), Rainfall (2 year lag, −), Pine beetle availability (+) |
| Site-level food availability and species interactions (competition, predation) | Availability of food for nestlings (breeding season) increases fecundity, but high densities of conspecific competitors and nest predators lower fecundity through delayed laying, clutch size reduction and nest failure[e] | DFE ∼ Chickadee density (+), Squirrel density (+) CS/NS/BS ∼ Lepidopteran availability (+), Pine beetle availability (+), Chickadee density (−), Squirrel density (−) |
| Nest-level | Earlier laying dates and larger cavities permit larger clutch sizes[f]. A larger clutch increases the chances of a nest surviving multiple partial-predation events[g]. Clutch size and nest survival may also be positively linked through correlations with unmeasured variables: they may both increase with parental condition[*h,i] and decline with predation risk[j]. Brood size (number of young fledged from successful nests) is directly constrained by clutch size. Smaller, higher cavities provide protection from predators, increasing nest survival and brood size. | CS ∼ DFE (−), cavity size (+) NS ∼ CS (+), cavity size (−), cavity height (+) BS ∼ CS (+), cavity size (−), cavity height (+) |

**Notes.**
[*]Factors not measured directly in this study.

In the past 120 years, this region sustained the highest number of mountain pine beetle outbreaks of any region in western Canada (*Taylor et al., 2006*). The study area comprised mixed coniferous-broadleaf forest with quaking aspen (*Populus tremuloides*), Douglas-fir (*Pseudotsuga menziesii*), lodgepole pine (*Pinus contorta*), and white and hybrid spruce *Picea glauca x engelmannii* (*Meidinger & Pojar, 1991*). Mountain chickadees used quaking aspen trees for 98% of nesting attempts (638 out of 654 nests found between 1995–2011; K Martin, 2011, unpublished data). Fourteen of the study sites were located near Riske Creek, 40 km west of Williams Lake (52°14′N, 122°12′W; where the weather station is located), and 9 sites were near Knife Creek, 20–40 km east of Williams Lake. Study sites ranged from 7 to 32 ha in size, and varied in composition from continuous forest to five sites that comprised several 'forest groves' (0.2 to 5 ha) within a grassland matrix. Additional study area details are given by *Martin & Eadie (1999)* and *Aitken, Wiebe & Martin (2002)*.

## Nest monitoring

We followed all protocols recommended by the Animal Care Committees of the University of British Columbia and Environment and Climate Change Canada, in accordance with annually renewed permits under the University of British Columbia Animal Care Protocol and Environment and Climate Change Canada's scientific permit and banding permit

**Table 2 Models in four sets, predicting fecundity of mountain chickadees at the three spatial scales.** Model selection results for mountain chickadee fecundity response variables (Response) for models compared using an information theoretic approach. For each model set, we list the response variable and sample size (n; number of nests). For each model, we report the scale, model name, fixed effects, ΔAICc (difference in AICc between the indicated model and the lowest AICc model), and Akaike weight (Wi). Models with ΔAICc < 2 are highlighted in bold and considered to be the most plausible models in the set. Fixed effects are highlighted in bold when the 95% confidence intervals on their parameter estimates did not overlap 0, and we indicate the direction of their effect on the response variable (+ or −). Models predicting laying date, clutch size, and brood size included a random effect of year.

| Response | n | Scale | Model | Fixed effects | ΔAICc | Wi |
|---|---|---|---|---|---|---|
| Laying date | 115 | Region | **Climate** | Snow depth, **Spring temperature** (−), Winter storms, **Spring temperature*Winter storms** (+) | 0 | 1 |
| | | | Winter food | Pine beetle, 2-yr-temperature, 2-yr-rain | 15.42 | 0 |
| | | | Null | ~ | 21.37 | 0 |
| | | Site | Null | ~ | 0 | 0.609 |
| | | | Predators & competitors | Squirrels, Chickadees | 0.89 | 0.391 |
| Clutch size | 78 | Region | **Null** | ~ | 0 | 0.874 |
| | | | Climate | Snow depth, Spring temperature, Winter storms, **Spring temperature*Winter storms** (−) | 4.69 | 0.084 |
| | | | Winter food | Pine beetle, 2-yr-temperature, 2-yr-rain | 6.06 | 0.042 |
| | | Site | **Null** | ~ | 0 | 0.936 |
| | | | Breeding season food, predators & competitors | Lepidopteran defoliators, Pine beetle, Squirrels, Chickadees | 5.37 | 0.064 |
| | | Nest | **Nest** | **Laying date** (−), Cavity height, Cavity size | 0 | 0.999 |
| | | | Null | ~ | 15.18 | 0.001 |
| Nest survival | 101 | Region | **Winter food** | Pine beetle, **2-yr-temperature** (+), 2-yr-rain | 0 | 0.75 |
| | | | Null | ~ | 3.08 | 0.161 |
| | | | Climate | Snow depth, Spring temperature, Spring storms, **Spring temperature*Spring storms** (−) | 4.24 | 0.09 |
| | | Site | **Null** | ~ | 0 | 0.622 |
| | | | Breeding season food, predators & competitors | Lepidopteran defoliators, Pine beetle, Squirrels, Chickadees | 1 | 0.378 |
| | | Nest | **Nest** | **Clutch size** (+), Cavity height, **Cavity size** (−) | 0 | 0.945 |
| | | | Null | ~ | 5.69 | 0.055 |
| Brood size | 77 | Region | **Null** | ~ | 0 | 0.934 |
| | | | Winter food | Pine beetle, 2-yr-temperature, 2-yr-rain | 6.37 | 0.039 |

*(continued on next page)*

**Table 2** (*continued*)

| Response | n | Scale | Model | Fixed effects | ΔAICc | Wi |
|----------|---|-------|-------|---------------|-------|-----|
| | | | Climate | Snow depth, Spring temperature, Spring storms, Spring temperature*Spring storms | 7.03 | 0.028 |
| | | Site | Breeding season food, predators & competitors | Lepidopteran defoliators, Pine beetle, **Squirrels** (−), Chickadees | 0 | 0.576 |
| | | | Null | ~ | 0.61 | 0.424 |
| | | Nest | **Nest** | **Clutch size** (+), Cavity height, Cavity size | 0 | 1 |
| | | | Null | ~ | 43.59 | 0 |

number 10365. To find nests, we observed breeding behaviors of adult mountain chickadees and all other cavity-nesting species, and visually inspected tree cavities previously occupied by any species of bird or mammal. We noted which species excavated the nest cavity, and then every 4–7 days we monitored all accessible nests in tree cavities using a video camera cavity monitoring system (for cavities up to 15 m above ground), or a mirror and halogen flashlight from a ladder (up to 6.5 m). We confirmed active nests by the presence of eggs or nestlings. We recorded the date of first egg laid (laying date), clutch size, nest fate (failed or fledged at least one nestling), and brood size (number of young just prior to fledging). We considered clutches complete when we observed the same number of eggs on at least two visits. Nests were deemed successful if one or more chicks were observed to fledge, or were still in the nest at least 17 days after hatching (minimum nestling period for mountain chickadee; *McCallum, Grundel & Dahlsten, 2020*), with no signs of predation after our last visit with large nestlings. We noted any sign of predation attempts (predator usurped cavity, cavity torn open, or nest material pulled out of the cavity).

We considered second clutches to be nests initiated after either failed attempts, or successful fledging of first broods (*Lack, 1954*). We were able to confirm a few cases of double brooding (a second brood following a successful first brood, all in 2005 and 2007) by color-banding 115 adults on breeding territories. These adults were captured using mist nets or by covering the nest entrance with a dip net when the adult was inside, banded using a unique combination of three plastic color bands and one numbered aluminium band, and released at the capture site within 10 min. Our field observations and an initial inspection of the data showed a clear bimodal distribution of laying date in 2005 and 2007 (but not in other years). There was a gap in laying around 3 June in both 2005 and 2007, so we considered all clutches laid after 3 June in those years as probable second clutches, and we omitted them from analyses of factors influencing laying date, clutch size, nest survival, and brood size for first nests. In 2002, laying was generally very late (earliest = 21 May, mean = 31 May), and in this year there were 5 clutches laid after 3 June (range 4–26 June) that we classified as probable first clutches.

## Nest cavity characteristics
When nesting was finished, we measured cavity height above ground using measuring tape or a 15-m measuring pole, and entrance diameter when it was possible to access

**Table 3 Parameter estimates for regional-, site- and nest-level generalized linear models explaining variation in mountain chickadee fecundity.** Fixed effects with a *p*-value <0.05 and in bold explained a significant amount of variation relative to other fixed effects examined within each model. Sample sizes vary according to data availability for response variables: relative laying date ($n = 115$ nests), clutch size ($n = 78$), nest survival ($n = 101$), and brood size ($n = 77$ nests). Direction of parameter estimates (b) indicates positive (no symbol) or negative (−) effects on response variables. Models predicting laying date, clutch size, and brood size included a random effect of year. Bold highlighting indicates plausible models ($\Delta AICc < 2$) that explained the data better than the null model.

| Response | *n* | Scale | Model | Fixed effects | Value | Std. Error | *t*- or *z*- value | *p*-value |
|---|---|---|---|---|---|---|---|---|
| Laying date | 115 | Region | **Climate** | Intercept | 140.63 | 0.61 | 230.19 | 0 |
| | | | | **Spring temperature** | −5.98 | 0.70 | −8.51 | **0** |
| | | | | Winter storms | 0.47 | 0.64 | 0.73 | 0.487 |
| | | | | Snow depth | 0.99 | 0.68 | 1.46 | 0.188 |
| | | | | **Spring temperature\* Winter storms** | 2.47 | 0.83 | 2.98 | **0.021** |
| | | | Winter food | Intercept | 140.17 | 1.64 | 85.43 | 0.000 |
| | | | | 2-yr-temperature | −1.76 | 1.39 | −1.27 | 0.241 |
| | | | | 2-yr-rain | −0.42 | 1.72 | −0.24 | 0.814 |
| | | | | Pine beetle | −2.79 | 1.82 | −1.53 | 0.164 |
| | | Site | Predators & competitors | Intercept | 141.00 | 1.67 | 84.56 | 0 |
| | | | | Squirrels | 0.70 | 0.78 | 0.90 | 0.370 |
| | | | | Chickadees | 0.14 | 0.73 | 0.19 | 0.849 |
| Clutch size | 78 | Region | Climate | Intercept | 1.88 | 0.03 | 60.20 | 0 |
| | | | | Spring temperature | 0.04 | 0.04 | 1.06 | 0.290 |
| | | | | Winter storms | −0.03 | 0.03 | −0.95 | 0.342 |
| | | | | Snow depth | 0.00 | 0.03 | 0.14 | 0.890 |
| | | | | **Spring temperature\* Winter storms** | −0.09 | 0.04 | −2.20 | **0.028** |
| | | | Winter food | Intercept | 1.88 | 0.04 | 46.35 | 0 |
| | | | | 2-yr-temperature | 0.03 | 0.04 | 0.76 | 0.447 |
| | | | | 2-yr-rain | 0.02 | 0.04 | 0.57 | 0.569 |
| | | | | Pine beetle | 0.02 | 0.05 | 0.43 | 0.666 |
| | | Site | Breeding season food, predators & competitors | Intercept | 1.87 | 0.04 | 48.21 | 0 |
| | | | | Lepidopteran defoliators | −0.02 | 0.02 | −0.99 | 0.323 |
| | | | | Pine beetle | 0.01 | 0.02 | 0.58 | 0.562 |
| | | | | Chickadees | 0.00 | 0.02 | 0.21 | 0.831 |
| | | | | Squirrels | −0.02 | 0.03 | −0.91 | 0.361 |
| | | Nest | **Nest** | Intercept | 1.88 | 0.04 | 47.48 | 0 |
| | | | | **Laying date** | −0.12 | 0.02 | −4.96 | **0** |
| | | | | Cavity height | 0.01 | 0.02 | 0.40 | 0.692 |
| | | | | Cavity size | −0.02 | 0.03 | −0.56 | 0.575 |
| Nest survival | 101 | Region | Climate | Intercept | 4.99 | 0.29 | 17.15 | 0 |
| | | | | Spring temperature | 0.11 | 0.34 | 0.32 | 0.750 |

| Response | n | Scale | Model | Fixed effects | Value | Std. Error | t- or z-value | p-value |
|---|---|---|---|---|---|---|---|---|
| | | | | Snow depth | −0.49 | 0.37 | −1.33 | 0.184 |
| | | | | Spring storms | −0.11 | 0.28 | −0.39 | 0.699 |
| | | | | **Spring temperature* spring storms** | −0.69 | 0.34 | −2.04 | **0.041** |
| | | | **Winter food** | Intercept | 4.92 | 0.28 | 17.52 | 0 |
| | | | | **2-yr-temperature** | 0.61 | 0.28 | 2.20 | **0.028** |
| | | | | 2-yr-rain | 0.16 | 0.29 | 0.55 | 0.581 |
| | | | | Pine beetle | 0.49 | 0.31 | 1.57 | 0.116 |
| | | Site | Breeding season food, predators & competitors | Intercept | 4.85 | 0.26 | 18.71 | 0 |
| | | | | Squirrels | −0.34 | 0.20 | −1.70 | 0.089 |
| | | | | Lepidopteran defoliators | −0.35 | 0.23 | −1.52 | 0.128 |
| | | | | Pine beetle | 0.14 | 0.33 | 0.42 | 0.675 |
| | | | | Chickadees | −0.11 | 0.25 | −0.46 | 0.647 |
| | | Nest | **Nest** | Intercept | 5.62 | 0.49 | 11.36 | 0 |
| | | | | **Clutch size** | 0.54 | 0.23 | 2.37 | **0.018** |
| | | | | Cavity height | −0.27 | 0.26 | −1.02 | 0.308 |
| | | | | **Cavity size** | −1.25 | 0.56 | −2.22 | **0.027** |
| Brood size | 77 | Region | Climate | Intercept | 1.76 | 0.04 | 39.96 | 0 |
| | | | | Spring temperature | 0.02 | 0.05 | 0.43 | 0.669 |
| | | | | Spring storms | 0.08 | 0.06 | 1.29 | 0.196 |
| | | | | Snow depth | 0.00 | 0.04 | 0.10 | 0.924 |
| | | | | Spring temperature* Spring storms | 0.06 | 0.05 | 1.28 | 0.199 |
| | | | Winter food | Intercept | 1.73 | 0.05 | 37.31 | 0 |
| | | | | 2-yr-temperature | 0.03 | 0.04 | 0.69 | 0.489 |
| | | | | 2-yr-rain | 0.02 | 0.05 | 0.32 | 0.749 |
| | | | | Pine beetle | 0.01 | 0.06 | 0.13 | 0.893 |
| | | Site | Breeding season food, predators & competitors | Intercept | 1.74 | 0.04 | 49.01 | 0 |
| | | | | Lepidopteran defoliators | 0.05 | 0.03 | 1.71 | 0.087 |
| | | | | Chickadees | −0.03 | 0.03 | −0.95 | 0.340 |
| | | | | **Squirrels** | −0.07 | 0.03 | −2.18 | **0.030** |
| | | | | Pine beetle | 0.06 | 0.03 | 1.85 | 0.064 |
| | | Nest | **Nest** | Intercept | 1.70 | 0.03 | 53.74 | 0 |
| | | | | **Clutch size** | 0.21 | 0.02 | 8.74 | **0** |
| | | | | Cavity height | 0.02 | 0.02 | 0.90 | 0.370 |
| | | | | Cavity size | 0.08 | 0.05 | 1.71 | 0.088 |

the cavity using a ladder. Following the criteria of *Edworthy et al. (2018)*, we classified cavity size as "small" if the entrance measured <3.5 cm in width or if we had observed that the cavity was excavated by a downy woodpecker (*Dryobates pubescens*), red-breasted nuthatch, black-capped chickadee (*Poecile atricapillus*), or mountain chickadee (rare, but did occur at 3 cavities that were used for 10 nesting attempts; out of 115 cavities used in 196 nesting attempts with known cavity origin). Cavities were classified as "medium–large" if the entrance measured >3.5 cm in width or if the cavity was excavated by a larger woodpecker (northern flicker; *Colaptes auratus*, red-naped sapsucker; *Sphyrapicus nuchalis*, hairy woodpecker; *Leuconotopicus villosus*, or American three-toed woodpecker; *Picoides dorsalis*).

## Predators and competitors data

To examine site-level factors influencing mountain chickadee fecundity, we established point count stations across our 23 sites (7–32 stations site$^{-1}$). In continuous forest sites, stations were spaced every 100 m in a grid $\geq$ 50 m from a grassland or wetland edge (one station ha$^{-1}$). In forest groves, where it was not possible to establish a grid, we placed point count stations at least 100 m apart. A total of 383 stations were surveyed for mountain chickadees, squirrels, and trees (we collected tree data in 0.04-ha circular plots centered at each point count station; see below); 295 stations were surveyed every year, and 88 had missing data for one or more years. To estimate relative population densities of mountain chickadee and red squirrel, each year from 2000 to 2011 between May and July we conducted 3 rounds of point counts at each point count station. During each 6-minute point count between 0500 and 0930 h, we recorded the species and number of individual chickadees and squirrels seen or heard within a 50-m radius of the station. We detected most individuals by vocalizations, and mapped each individual's location relative to the centre of the point count station to avoid double counting. To address detection bias by observers across counts, we alternated observers across sites and paired each observer with one of the two principal investigators at least once per season. While it is possible that 6-minute counts could lead to an underestimate of true densities, detection probability was maximised by sampling at point count stations that were typically spaced only 100 m apart. Also, we examined the effect of changes in densities of squirrels and chickadees on fecundity and were therefore concerned only with relative densities and not absolute densities at each site in each year. Further details on point count surveys are provided in *Martin & Eadie (1999)* and *Drever et al. (2008)*. To generate a measure of squirrel and mountain chickadee density (detections per point count) for each year of the study at each of our 23 sites, we first took the mean number of individuals detected across the three counts at each station, then took the mean across all stations within the site and year. We used a Pearson's correlation coefficient to examine the relationship between mountain chickadee density from point counts, and the number of nests found in each year (excluding probable second nests).

## Prey availability data (site-scale and regional)

Mountain chickadees are opportunistic feeders: in autumn and winter, they live in social groups, feeding on arthropods and conifer seeds (which they cache); in spring and summer,

they are socially monogamous, territorial, and feed on insects (including Coleoptera, Lepidoptera, Hymenoptera, and Homoptera), mostly in conifers (*McCallum, Grundel & Dahlsten, 2020*). Nestlings are fed arthropods (including adults and larvae of mountain pine beetle and western spruce budworm), and at the peak of nestling food demand, adults preferentially take the most readily available prey (*Grundel & Dahlsten, 1991*; *Grundel, 1992*; *McCallum, Grundel & Dahlsten, 2020*). Mountain chickadees exhibit a strong functional response to insect outbreaks, with spruce beetles (Coleoptera: *Dendroctonus obesus*) comprising approximately 27% of adult diet under outbreak conditions in California, and dozens to hundreds of lodgepole needle miners (Lepidoptera: *Coleotechnites milleri*) found in chickadee stomach contents under outbreak conditions in Colorado (*Telford & Herman, 1963*; *Baldwin, 1968*). In a nest box study in California, they laid larger clutches in years with higher proportions of arthropods (*vs.* plant material and grit) in the diet prior to the breeding season (*Dahlsten & Copper, 1979*). Because mountain chickadees forage widely in winter (when they likely accumulate resources to breed, as is the case for a close congener, black-capped chickadee; (*Montreuil-Spencer et al., 2019*)) but close to the nest in spring (when they begin breeding; *McCallum, Grundel & Dahlsten, 2020*), we examined prey availability at the regional scale in winter and at the site level in spring. Mountain pine beetle larvae are available to mountain chickadees year-round and were included in both regional and site-level analyses; lepidopteran larvae are only available in spring and summer, and were therefore included only in site-level analyses. To estimate prey availability we used a combination of tree surveys on the ground, and data on the extent of lepidopteran defoliation from the British Columbia Ministry of Forests and Range (*Westfall, 2007*).

An outbreak of mountain pine beetle occurred across all sites in our study area, with the incidence of attacked pines increasing sharply after 2002; over 95% of the mature lodgepole pine trees (40% of the trees on our sites) were dead by 2005 (*Drever, Goheen & Martin, 2009*; *Edworthy, Drever & Martin, 2011*). However, the onset and peak years of the beetle outbreak varied across sites (*Drever, Goheen & Martin, 2009*; *Norris & Martin, 2014*). In British Columbia, mountain pine beetles lay their eggs in late summer beneath the bark of living pines, and larvae develop under the bark through the winter and following spring, emerging as adults in summer (*Reid, 1962*). Outbreaks occur in susceptible forest stands when climatic conditions become favorable for survival of adults and larvae; specifically, a warm August to promote flight by breeding adults followed by a warm winter to enhance survival of larvae, and then low precipitation the next spring (*Safranyik, 1978*; *Safranyik & Carroll, 2006*). Beetle larvae available to chickadees during the breeding season were evident the previous summer by the presence of dried resin outflows, or small entry holes (~2 mm in diameter) in the bark. Each summer, we assessed beetle-infected pine densities for the following spring (chickadee breeding season) by counting the total number of live pine trees ≥ 12.5 cm diameter at breast height (dbh), with bark boring insects, in the 0.04-ha plots centered at each point count station. We subtracted any trees that fell or were cut over the winter, then divided the remaining total by 0.04, and divided again by the number of plots at the site to estimate beetle-infected pines ha$^{-1}$ at the site level (prey available to mountain chickadees during the breeding season). To assess regional-level mountain pine

beetle (prey available to mountain chickadees in each winter), we took the mean number of beetle-infected pines ha$^{-1}$ across the study area, including only the 295 stations that were surveyed in all years.

Defoliating lepidopterans in our study area include western spruce budworm, which affects living Douglas-fir and spruce, and aspen leaf miner (*Phyllocnistis populiella*), northern tent caterpillar (*Malacosoma californicum pluviale*), forest tent caterpillar (*Malacosoma disstria*), large aspen tortrix (*Choristoneura conflictana*), and satin moth (*Leucoma salicis*), which affect aspen (*Finck, Humphreys & Hawkins, 1989*). Each of these defoliators lay eggs in mid-summer; eggs or early instars overwinter on living leaves or stems/branches of their host tree, and then emerge as larvae in the next spring to summer to feed on the buds and/or new foliage until transforming to adults in mid-late summer (*Finck, Humphreys & Hawkins, 1989*). Peak outbreak conditions occurred in 2007 for western spruce budworm, and in 2002 for lepidopterans affecting aspen. Under peak conditions, nearly 100% of living host trees at our study area were attacked. To assess lepidopteran density, we combined our field inventory of trees with regional-scale aerial overview survey data for the Cariboo Forest District (excluding Quesnel; *BC Ministry of Forests, 1996*). We first calculated the density of host trees for each site and year (sum of counts of trees in 0.04-ha circular plots at our point count stations, divided by 0.04, and divided again by the number of plots at the site to give trees ha$^{-1}$). We used density of living Douglas-fir and spruce trees for western spruce budworm, and living aspen for the remaining defoliators. We then obtained an index of outbreak intensity for each year, from the aerial overview data, by dividing the area defoliated that year by the maximum area defoliated at the peak of the respective outbreak (2002 for aspen defoliators, 2007 for western spruce budworm). We multiplied these indices by the densities of host trees in each site and year to estimate infected trees ha$^{-1}$ at each site each year for western spruce budworm and aspen defoliators, respectively. Finally, for each site-year combination, we added the estimated density of trees infected by western spruce budworm to the estimated density of trees infected by aspen defoliators, for a total estimated density of lepidopteran-infected trees at each site during the breeding season of each year.

## Climate data (regional scale)

Climate measurements were inferred across the study area for each year, by accessing data from the Environment and Climate Change Canada weather station Williams Lake A (WMO ID 71104; 52°10′48″N, 122°03′00″W; elevation 939.7 m; http://climate.weather.gc.ca). We calculated spring temperature as the mean of daily maxima across the period from 7 March to 20 May (the best time window for predicting laying phenology of mountain chickadees, according to *Drake & Martin, 2018*). Snow depth prior to mountain chickadee breeding was calculated as mean snow depth from 1 January to 30 April. We calculated the number of winter storms (expected to influence laying date and clutch size) as the number of days with rain and/or snow (precipitation) >10 mm between 1 January and 30 April. We calculated the number of spring storms (expected to influence nest survival and brood size) as the number of days with precipitation >10 mm between 1 May and 30 June.
We examined climate variables associated with conifer seed production because mountain chickadees often forage on conifer seeds over winter (*McCallum, 1990*; *McCallum, Grundel & Dahlsten, 2020*), and increases in winter food availability can lead to earlier laying dates (*e.g.*, *Smith et al., 1980*) and increased clutch size in passerine birds (*e.g.*, *Broggi et al., 2022*). Conifer seed production in dry mixed conifer forests of western North America (in fall) is highly correlated with up to a one-year lag in growing season temperature and soil moisture (*Rother, Veblen & Furman, 2015*; *Petrie et al., 2016*); therefore, we calculated 2-year lagged temperature and rainfall for use as predictors in models of mountain chickadee fecundity. We considered rainfall instead of total precipitation because rainfall increases annual soil moisture more consistently than snow due to annual fluctuations in water retention/drainage during snowmelt (*Williams, McNamara & Chandler, 2009*). We considered 2-year lagged growing season temperature as the mean daily temperature from 1 April through 30 August, 2 years prior to the breeding attempt. Similarly, we calculated 2-year lagged rainfall as the sum of all rainfall from 1 September in year x-3 through 30 August in year x-2, where x corresponds to the year of the breeding attempt.

## Statistical analyses

We used mixed-effects models to test *a priori* hypotheses about the influence of regional, site-level, and nest-level factors on four components of mountain chickadee fecundity (response variables for each nest were: laying date, clutch size, nest survival, and brood size; Table 1 and Table S1). Probable second nests (first egg laid after day 150 in 2005 or after day 160 in 2007) were excluded from all models. We elected to run separate models predicting each component of fecundity (rather than a single path analysis) because many nests had incomplete fecundity data, resulting in different but overlapping datasets for each response variable. We assumed that the nests for which appropriate fecundity data were available were independent and representative of all nests present in each year. At the regional scale, we constructed two models for each fecundity response variable, and considered models to be plausible (supported by the data) if their ΔAICc values were within 2 values of the top model. At site- and nest-scales, we constructed one model for each fecundity response variable. At all scales, fixed effects were considered to have statistical support from the data if the 95% confidence intervals on parameter estimates did not overlap zero, and if they were included in a model with ΔAICc values >2 points below the null (intercept-only) model. We predicted that the number of storms would affect the correlation between annual fecundity and spring temperature such that winter storms would impede the ability of females to lay earlier and larger clutches in warmer springs, and spring storms would interfere with feeding nestlings leading to greater partial nest loss (*de Zwaan et al., 2020*). Therefore, we included interaction terms between spring temperature and storms for models explaining variation in each fecundity response variable.

All data analyses were conducted in R version 3.5.3 (*R Core Team, 2019*); packages listed in Table S1). Continuous predictors were scaled to have a mean of 0 and standard deviation of 1. Due to multiple measurements of fecundity within cavities (nests) and sites across years, the error term of nest and site crossed with year would have appropriately

reflected the structure of the data. However, including site as a random effect resulted in singularity in some models, likely as a result of overfitting. Therefore, we applied a model selection approach to choose the model with the random effects term that balanced predictive accuracy and overfitting/type I error (*Bates et al., 2015*; *Matuschek et al., 2017*). We excluded the random effect of site because where the models converged and did not result in singularity, a comparison between top models for laying date, clutch size and brood size, with random effects of (1) year only and (2) site crossed with year, the model with year only was better supported by the data ($\Delta$AICc <2). We lacked sufficient data to include a random error term to account for within-nest variation (at the nest scale), and therefore were unable to compare models across scales, so we limited our model selection approach to within scales only. For nest survival, some models with a random effect of year resulted in singularity, and removing the random effect resulted in very similar parameter estimates. Thus, we included only year as a random effect in models of laying date, clutch size, and brood size, and no random effects in models of nest survival. We examined the correlation coefficients of predictors using correlation matrices of fixed effects for each model, and found no significant correlations ($r < 0.7$). We modeled laying date (continuous response variable) using general linear mixed models with a Gaussian error structure, and examined residuals and Q-Q plots to check model fit and assumptions (*Pinheiro et al., 2021*). We modeled clutch size and brood size (under-dispersed counts), using generalized linear mixed models with COM-Poisson error structure to estimate a dispersion parameter to handle underdispersion (*Sellers, Borle & Shmueli, 2012*; *Brooks et al., 2017*). We modeled daily nest survival (binary) using the logistic exposure method described by *Shaffer (2004)*, a variation on logistic regression in which the link function accounts for the number of exposure days between observer visits to the nest. In modeling brood size, we restricted the dataset to nests that were successful. In modeling nest survival and brood size, clutch size was included as a predictor, so data were restricted to nests with complete clutches of eggs observed (difficult to obtain given the configuration of the nest cups within tree cavities and the chickadees' habit of covering their eggs before leaving the nest). Results are reported as mean ± standard deviation.

## RESULTS

### Annual variation in fecundity

In 12 years across 23 sites, we studied 513 mountain chickadee nests in tree cavities primarily excavated by woodpeckers and nuthatches. Incubation period was 13 ± 1 days ($n = 60$ nests) and nestling period was 19 ± 1 days ($n = 14$ nests). Initiation of egg laying ranged annually over a 10 week period (earliest clutch initiation was on 1 May in 2005 and latest on 9 July in 2007 (Fig. 2A). In most years, laying was finished by 9 June; however, in 2005 and 2007, there were two peaks in laying date—one in mid-May, and the second in late June. At least 32 of the late nesting attempts (and none of the early nesting attempts) were second clutches in a tree or vicinity where mountain chickadees had nested earlier in the season (confirmed by banding to be the same pair in six cases). Overall, mean laying date was 20 May for 129 known and probable first clutches, and 23 June for 32 known and

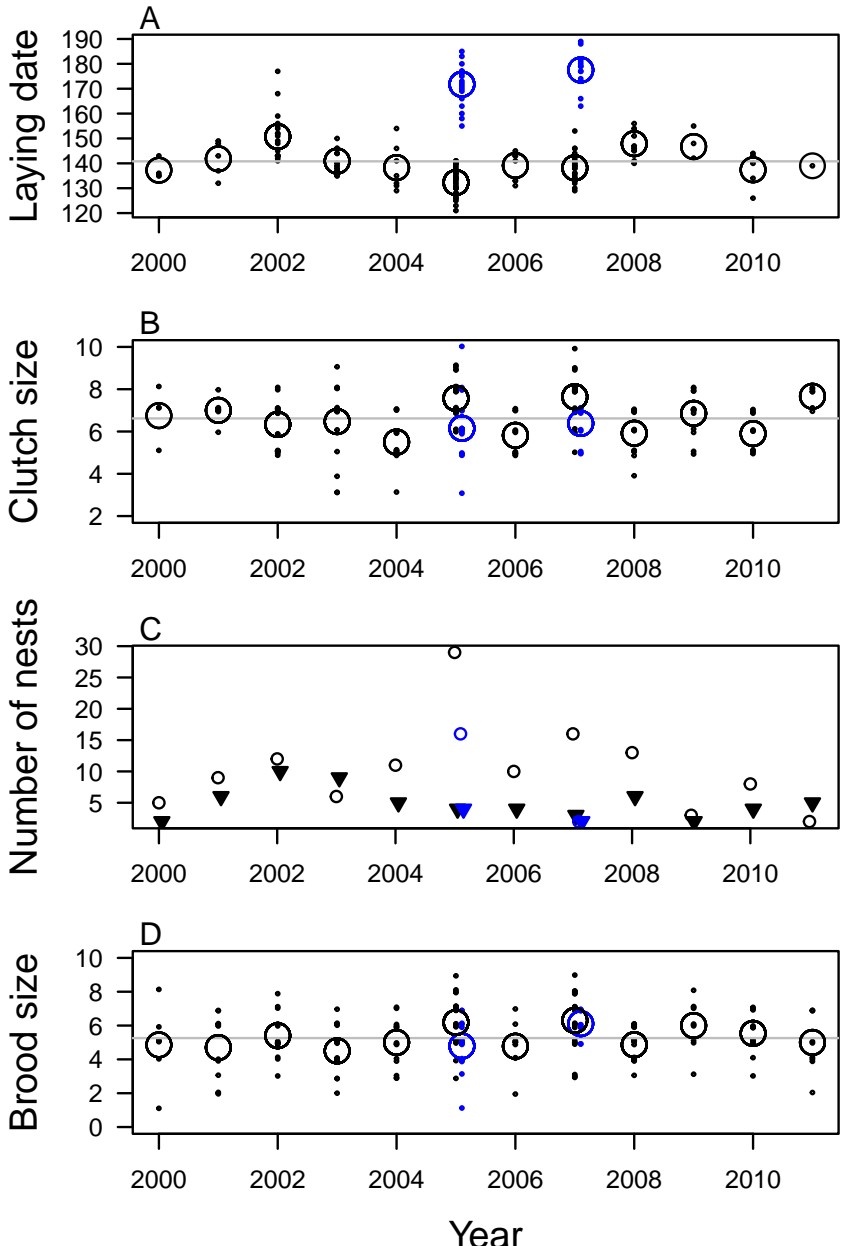

**Figure 2  Variation in fecundity of mountain chickadee from 2000 to 2011.** (A) laying date, (B) clutch size, (C) number of nests that fledged (open circles) and failed (filled triangles), (D) brood size (number of young fledged) for successful nests. Black indicates first clutches and blue indicates known and probable second clutches (found only in 2005 and 2007). For (A), (B), and (D) small points indicate individual nesting attempts; large open circles indicate annual means.

probable second clutches. Mean hatch day was 6 June ± 8 days for 45 known first clutches, and 32 days later, 8 July ± 7 days for 10 known second clutches. On two occasions, parents were observed feeding fledglings of the first brood while attending the second nest.

Clutch size was 6.81 ± 1.25 eggs (range: 3–10) for known and probable first clutches ($n = 189$; Fig. 2B). Known and probable second nests had a clutch size of 6.24 ± 1.23 eggs (range: 3–10) ($n = 33$). We determined nest fate for 208 nests, of which 142 (68%) fledged at least one nestling (67% of 184 first nests and 75% of 24 second nests); nest success was highest in 2005 (88% of 33 first nests, 80% of 20 second nests) and lowest in 2011 (29% of 7 nests) and 2003 (40% of 15 nests; Fig. 2C). We found evidence of predation at 45 of the 66 failed nests (nest was empty before 14 days post-hatch and the cavity was usurped by a predator, torn open, or nest material removed). At successful nests, mean brood size of first nests was 5.48 ± 1.55 fledglings (range: 1–9) and varied annually, with mean brood size highest in 2007 (6.30) and 2005 (6.18) and lowest in 2003 (4.50; Fig. 2D). Mean brood size for second nests was 5.21 ± 1.37 fledglings (range: 1–7). The total number of probable first nests found in each year was positively correlated with the annual mean number of mountain chickadee individuals detected per ha in point counts ($r = 0.70$, $df = 10$, $p = 0.01$).

### Factors influencing fecundity in first nests

Models at the regional scale explained some of the variation in laying date and nest survival of mountain chickadees; models at site scale (insect availability, conspecifics, predators) performed poorly at explaining variation in fecundity (no better than the null model); and models at nest scale explained some of the variation in clutch size, nest survival, and brood size (Table 2, Fig. 1). Earlier laying was associated with warmer spring temperatures; however, this relationship was reduced with increasing number of storms from Jan 1 through Apr 30 (Tables 2 and 3; Figs. 1 and 3). Clutch size declined with laying date both within and across years (Tables 2 and 3; Figs. 1 and 4). Daily nest survival (DSR) increased with annual temperature 2 years prior to the nesting attempt at the regional scale, and increased with clutch size but declined with cavity size at the nest scale (Tables 2 and 3; Figs. 1 and 5). Extended over the 39-day nesting period from clutch initiation to fledging, probability of nest survival (DSR$^{39}$) ranged from 21% in a large cavity with a clutch size of 4, to 95% in a small cavity with a clutch size of 9, and was 88% for the mean clutch size of 6.81 in (typical) small cavities. Brood size (in successful nests), was strongly and positively predicted by clutch size only, and year effects were negligible in this relationship (Tables 2 and 3; Figs. 1 and 6).

Three models included statistically significant effects related to climate and predators, but were no better than the null models at explaining variation in fecundity. In the regional climate models, clutch size and nest survival increased with spring temperature except when there were more winter storms (Tables 2 and 3). At the site level, brood size declined with increasing squirrel density (Table 3). None of the chickadee fecundity measures were predicted by our measures of conspecific densities, cavity height, or snow depth in any of the models (Table 3).

## DISCUSSION

During our 12 year study, as weather conditions varied, insect outbreaks increased food availability, and mountain chickadee abundance doubled, we found considerable variation

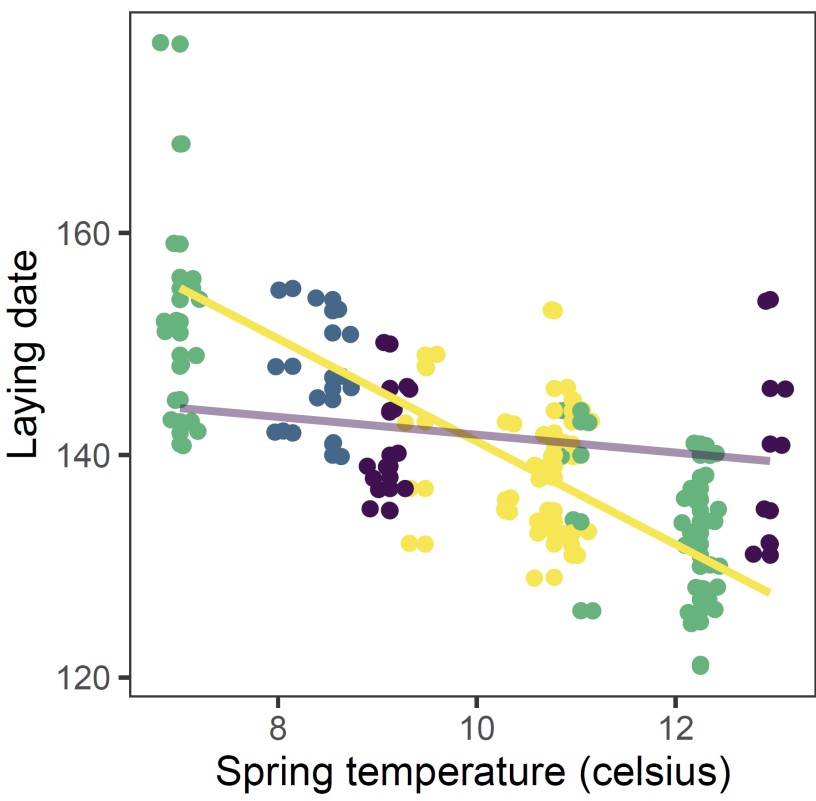

**Figure 3** Observed (points) and predicted (lines) laying date of first nests of mountain chickadee *vs.* spring temperature in 2000 to 2011. Points are slightly offset on both axes to avoid overlap. Predictions are based on the top (regional-scale climate) model in the set predicting laying date (Table 2), holding snow depth at its median (19.7 cm) and year at 2011. Colours indicate the number of storms from January to April: yellow = 0 storms, green = 1 storm, blue = 2 storms, purple = 3 storms. Straight lines indicate the predicted values of the regional climate model, in which there was a significant interaction between spring temperature and number of storms.

in laying date, clutch size, and production of chickadee fledglings in natural tree cavities. However, we did not find strong evidence linking this variation in fecundity to bottom-up (prey availability) or top down (*via* conspecific/predator densities) processes at the site scale. Laying date could only be linked to climate, with warmer springs resulting in earlier laying, as long as there were few storms from January to April. Mountain chickadees laid clutches of three to 10 eggs, with larger clutches when laying date was earlier, associated with warmer springs (in the absence of storms). As expected, clutch size correlated positively with both nest survival and brood size (*i.e.*, nest-level productivity), suggesting that clutch size is a good measure of mountain chickadee fecundity. Nest survival increased in smaller cavities, and there was evidence for the hypotheses that winter food availability improved nest survival at the regional level, and weak evidence that nest predation by squirrels restricted brood size at the site level. In summary, we found strong annual variation in our fecundity measures for mountain chickadees that was best explained by weather variables at the regional level, and nest cavity and clutch sizes at the nest level.

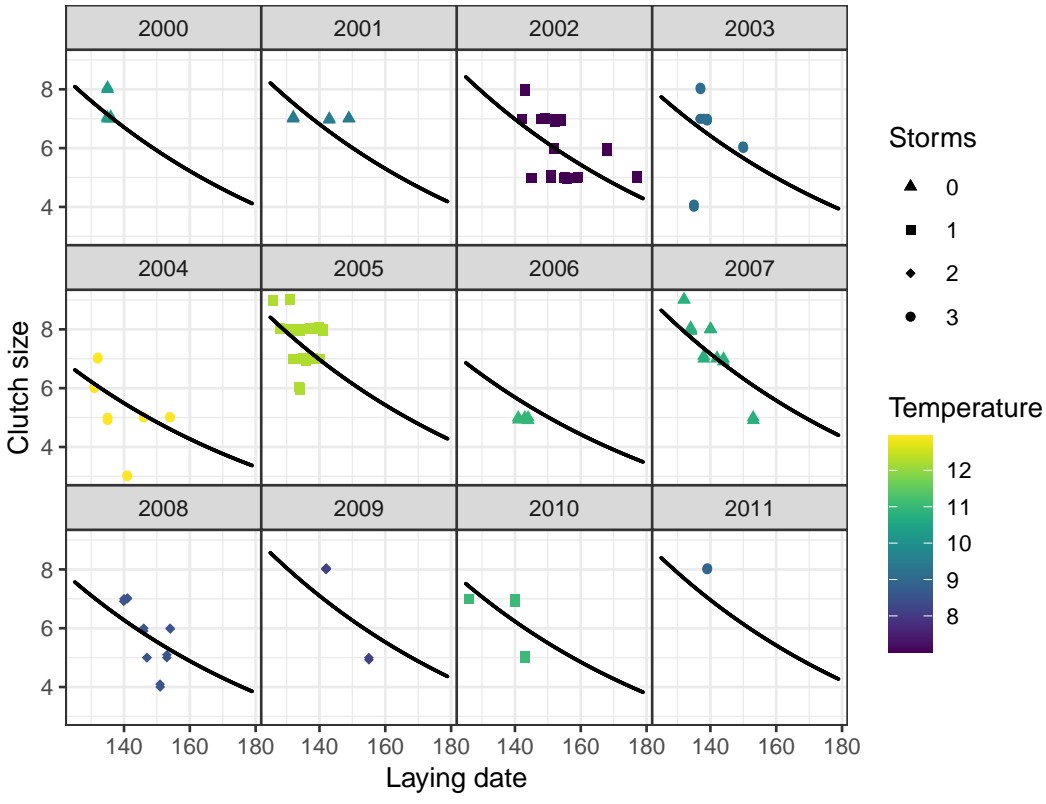

**Figure 4 Observed (points) and predicted (black lines) clutch size of first nests, *vs.* laying date for mountain chickadee, with the random effect of year (denoted by labels 2000 to 2011).** Predictions are based on the top (nest-scale) model in the set predicting clutch size (Table 2), holding cavity size as small, and cavity height at its median (3 m). Colors indicate spring temperature and symbols indicate the number of storms from January to April.

As predicted by our nest-level hypothesis, early laying dates were associated with larger clutches in mountain chickadees (and this relationship was observed both within, and across years). A relationship between clutch size and laying phenology has been established in many avian studies (*Visser, Holleman & Caro, 2009*; *Dunn, 2019*), but was not observed in mountain chickadees using nest boxes along a rural–urban gradient in interior British Columbia (*Marini et al., 2017*). Many hypotheses have been proposed to explain the seasonal decline in avian clutch size (*Decker, Conway & Fontaine, 2012*), including declines in food availability (*Perrins, 1965*), declines in offspring value (*Lepage, Gauthier & Menu, 2000*), time-limitation (*Siikamäki, 1998*), and earlier breeding by more experienced individuals and/or those in better condition (*e.g.*, possibly through access to more food over winter; *Hochachka, 1990*). Although we were unable to discriminate among these hypotheses using our dataset, we found no evidence that abundance of mountain pine beetle or lepidopteran defoliators during the breeding season influenced fecundity variables, nor that densities of conspecifics or predators (*i.e.*, squirrels) influenced relative

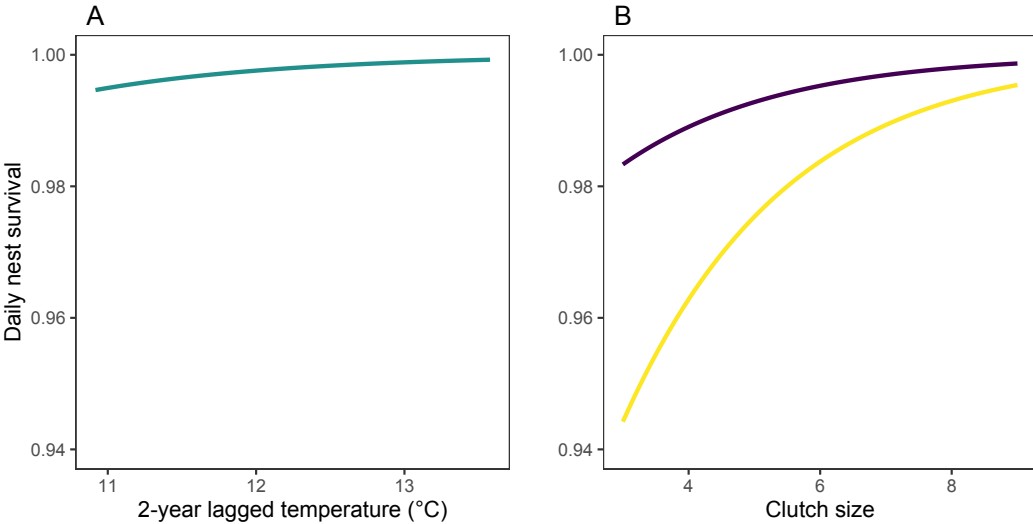

**Figure 5 Predicted values of daily nest survival rate (DSR) for mountain chickadee nests.** (A) Predicted values (turquoise) are derived from parameter estimates from the top regional model of daily nest survival (Table 3), allowing 2-year lagged temperature to vary between its minimum and maximum, while holding 2-year lagged rainfall at its median (306 mm) and study area-scale mountain pine beetle at its median (17 beetle-infected pines ha$^{-1}$). (B) Predicted values are derived from parameter estimates from the nest-scale model for small (purple) and large (yellow) tree cavities, allowing clutch size to vary between its minimum and maximum, and holding cavity height at its median (3 m).

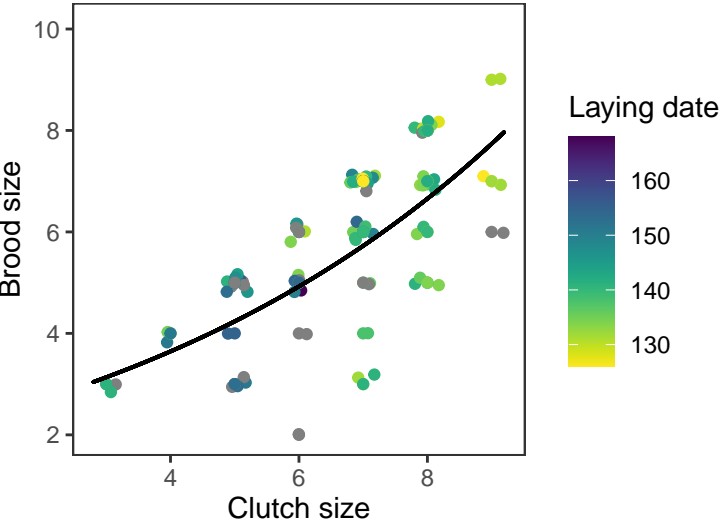

**Figure 6 Observed (points) and predicted (black line) brood size of first nests, *vs.* clutch size, for mountain chickadee.** Points are slightly offset on the *x* and *y* axes to show all data. Predictions are based on the top (nest-scale) model in the set predicting brood size (Table 2), holding cavity size as small, and cavity height at its median (3 m). Colors indicate laying date (date of first egg); grey points indicate 15 nests for which laying date was not known.

laying date, which suggests that the relationship between timing and clutch size was probably not related to territory quality.

Predation was the primary source of nest failure in our study population (at least 70% of known nest failures were caused by predation), as in most birds (*Ricklefs, 1969*; *Dahlsten & Copper, 1979*; *Martin, 1993*). Although we were unable to determine the predator species for most failed nests, red squirrel was the most abundant cavity-nesting mammal in our study area and is a common predator of the nests of chickadees and other songbirds (*Martin, 1993*; *Martin, Aitken & Wiebe, 2004*; *Mahon & Martin, 2006*). Although squirrel density was a negative predictor of brood size at the site level, the model in which it was included was no better than the null model, and it was not a good predictor of daily nest survival of mountain chickadees. Squirrel density explained a significant amount of variation in brood size in the red-breasted nuthatch in our study area (*Norris & Martin, 2014*), but did not predict nest survival in the chestnut-backed chickadee (*Poecile rufescens*) at another site in interior British Columbia, possibly because in mast years squirrels fed mostly on seeds, resulting in lower nest failure despite similar annual densities of squirrels (*Mahon & Martin, 2006*). In boreal forest, red squirrels foraged on bark beetles in years of spruce seed scarcity (*Pretzlaw et al., 2006*). Our finding that squirrel density was not a significant predictor of daily nest survival in mountain chickadee might be explained by squirrels foraging primarily on seeds and insects, relative to chickadee eggs and nestlings. In addition, chickadees might have assessed nest predation risk prior to breeding and reduced their reproductive investment (*i.e.*, fewer nests, smaller clutches) in years with higher densities of squirrels, even when weather conditions were favorable for earlier, larger, and more successful clutches (*e.g.*, 2004; *Norris & Martin, 2010*), thereby minimizing the negative effect of squirrel densities on fecundity and driving the positive correlation between nest survival and clutch size we found at the nest level. The negative relationship between cavity size and daily nest survival suggests that by selecting smaller cavities that excluded larger-bodied predators, as was shown in response to forest insect outbreaks (*Norris, Drever & Martin, 2013*; *Cockle & Martin, 2015*), mountain chickadees could increase their fecundity, which might have also countered the effects of squirrel densities on nest survival. Furthermore, the result of higher nest survival in smaller cavities provides a potential mechanistic explanation for the legacy effect of red-breasted nuthatch populations (*i.e.*, the positive correlation between mountain chickadee populations and a one-year lag in red-breasted nuthatch populations previously reported in *Norris, Drever & Martin, 2013*).

We found limited evidence to support the hypothesis that fecundity at individual nests of mountain chickadees is limited by food, even though in an earlier study at this site, the annual abundance of adult chickadees (from point counts) correlated with outbreaks of mountain pine beetle and western spruce budworm (*Norris, Drever & Martin, 2013*). Availability of mountain pine beetle and lepidopteran defoliators did not explain the variation in our fecundity variables. Moreover, although 2-year lagged temperature and rainfall were expected to increase conifer seed production over winter (*Owens, 2006*; *Rother, Veblen & Furman, 2015*; *Petrie et al., 2016*), we found only evidence relating 2-year lagged temperature to nest survival, and no evidence relating 2-year lagged rainfall to any

fecundity variables. It is possible that warmer temperatures 2 years prior to the breeding season contributed to conifer seed production in the intervening year (not measured), and that the increase in conifer seeds contributed to an increase in nest survival through improved winter body condition prior to breeding (*Montreuil-Spencer et al., 2019*). It is also quite possible that our indirect measures of conifer seed production (2-year lagged climate variables) did not adequately represent winter food supply. Future studies that measure directly the availability of all winter foods would be more appropriate to test the hypothesis that winter food supply drives changes in annual fecundity for mountain chickadees.

Beetle and budworm outbreaks may have combined with warm springs to increase annual fecundity by promoting double-brooding. Second broods following successful first broods are rare in chickadees due to their relatively long breeding season duration (in this study, 39 days from clutch initiation to fledging based on the mean clutch size of seven eggs). This is the first study, to our knowledge, to report mountain chickadees producing two successful clutches within the same season in tree cavities (*Foote et al., 2010*; *McCallum, Grundel & Dahlsten, 2020*). For species in which multiple brooding is common, pairs that initiate earlier tend to have a second brood, and food-supplemented birds often produce a second clutch in the season as a result of earlier initiation of first clutches (*Kluyver, 1951*; *Martin, 1987*; *Arcese & Smith, 1988*; *Nagy & Holmes, 2005*). In nest boxes in California, however, mountain chickadees laid second clutches in years with cool, wet springs, when first clutches were later and smaller relative to warmer springs, and the breeding season was extended into late summer (*Dahlsten & Copper, 1979*; *Dahlsten et al., 1992*). Our known second clutches occurred exclusively in warm years coinciding with peak abundance of mountain pine beetle (2005) and western spruce budworm (2007), when first clutches and broods were early and large, and densities of breeding pairs were high; in contrast, warm spring temperatures in 2004 did not result in more breeding pairs, large clutches, or second clutches, despite laying dates that were similar to 2007 (Fig. 2, *Drake & Martin, 2018*). We suspect that a greater number of storms during the breeding season (during and after laying) in 2004 prevented some pairs from breeding, and those that bred laid smaller clutches. Mountain pine beetles can produce their own second broods in warmer summers, when females lay their eggs under the bark of one tree, then re-emerge the same summer (instead of dying beneath the bark), and lay a second brood beneath the bark of another tree, further increasing the availability of insect prey in late summer (*Alfaro et al., 2003*). Moreover, earlier breeding of mountain chickadees in nest boxes in interior British Columbia was associated with earlier peak caterpillar abundance in urban (*vs.* rural) areas (*Hajdasz et al., 2019*), suggesting that earlier timing of food availability for nestlings (*e.g.*, due to warmer springs) is an important driver of earlier breeding in these mountain chickadee populations. It is possible that multiple factors, including high spring temperatures in years with few storms (early first nests) and high food availability (insect outbreaks that may have been extended in warmer years with double-brooding beetles) combined to promote the opportunistic double-brooding that we observed in mountain chickadees in 2005 and 2007.

Although we attempted to measure or estimate the factors most likely to influence annual fecundity of mountain chickadees in natural tree cavities, there are several important caveats to our study. As in any observational study, it is possible that the correlations (and lack of correlations) we found were driven by other, unmeasured variables. In particular, the lack of association between insect food availability and fecundity measures could indicate that food availability was not a strong and consistent driver of fecundity, but also that our measures of food availability were too coarse. Although we observed mountain chickadees feeding on mountain pine beetle and lepidopteran defoliators, and taking these items to their nestlings, we were unable to quantify the importance of these food items in their diet, and we lack precise measures of their availability at our study sites, especially for lepidopterans, which were estimated indirectly. Production of conifer seeds (particularly lodgepole pine), a key component of mountain chickadee diet in winter, may increase when trees are stressed by the same climate variables driving insect outbreaks, providing even more food than we estimated (additional seeds in winter, and insects year-round; *Safranyik, 1978*; *Rother, Veblen & Furman, 2015*; *Petrie et al., 2016*; *McCallum, Grundel & Dahlsten, 2020*). Ideally, measurements of fledgling physiology and diet would improve our understanding of how food, predators, and climate affect body condition and survival of fledglings.

Our study of mountain chickadees nesting under natural conditions, in cavities in decayed trees, showed strong variation in annual fecundity measures of these free-living small-bodied cavity-nesting birds. Our study conditions were not conducive to safely banding or taking measurements of nestlings, or marking and measuring most adults, to incorporate adult age, and adult and nestling condition, as additional correlates of fecundity. An understanding of this individual variation could be gained from studies in nest boxes, with the caution that nest boxes might obscure or conflate the effects of external drivers, for example by increasing cavity supply. Additionally, our regional-scale measures of climate and prey were uniform across all nests within a year. We accounted for the repeated sampling within years by including year as a random effect in our models, but the relatively small number of study years (12) may have prevented us from detecting or teasing apart the influence of some regional-scale variables.

Although we applied a longer-term observational approach to overcome some of the limits of short-term experimentation, our study questions and interpretations were nonetheless strongly influenced, and limited by, the values and concepts of a (reductionist) western scientific framework, under which we seek to understand variation by elucidating the drivers that likely vary according to the temporal and spatial scales of observation. To improve our understanding of the high annual variation in fecundity observed in mountain chickadee, we might look to Indigenous epistemologies that start from the assumption of interconnectedness, and can reveal patterns and processes overlooked in typical ecological studies (*Service et al., 2014*; *Henson et al., 2021*; *Tengö et al., 2021*). A co-production of knowledge approach collates Indigenous and western sciences to generate new understandings of the world that would not be achieved through the application of only one knowledge system (*Henri et al., 2021*; *Yua et al., 2022*). Future work on opportunistic

bird species, co-produced with Indigenous scientists and communities (*e.g.*, *Westwood et al., 2020*), could reveal a better understanding of the resiliency of wildlife species to climate change, and help guide future forest management and conservation efforts for climate-sensitive species.

## CONCLUSIONS

Our 12-year study in a natural tree-cavity system revealed strong variation in annual fecundity of mountain chickadees. Mountain chickadees advanced their laying date in warmer years (as shown by *Drake & Martin, 2018*), but only when there were few storms. The years of highest productivity (double brooding, large clutches, and a greater number of successful nests with large broods in smaller cavities) were years with warm springs and few storms, at the peak of mountain pine beetle (2005) and western spruce budworm outbreaks (2007). We can predict that warmer temperatures and associated insect outbreaks, which increase the populations of small-bodied excavators, may promote earlier breeding, larger, and more successful clutches and broods, and more frequent double-brooding in mountain chickadees, as long as the number of storms before and during breeding does not increase. However, variation in chickadee fecundity could not be fully explained by our measures of regional climate, regional or site-level food, or site-level densities of conspecifics or predators. We stress the importance of a study framework that includes long-term monitoring of fecundity, intra- and interspecific interactions, and food availability across variation in climate, to identify the mechanisms by which climate change might affect the phenology, annual fecundity, and recruitment of resident forest birds (*Pearce-Higgins et al., 2015*; *Halupka & Halupka, 2017*).

## ACKNOWLEDGEMENTS

This study was conducted on the lands of the Tŝilhqot'in, Secwépemc, and Southern Dakelh Peoples in an area also known as the Cariboo-Chilcotin region of British Columbia. The authors wish to acknowledge that Indigenous Peoples have been acquiring knowledge of and managing these ecosystems long before European colonization and western science was conducted in North America. We thank numerous graduate students and field assistants, particularly K. Aitken, M. Mossop, D. Gunawardana, M. Marjanovic, I. Behret, H. Kenyon, M. Edworthy, M. Behret, P. Robinson, and K. Scotton. D. Cockle designed, built and maintained the cavity monitoring equipment. The manuscript was improved by discussions with J. Goheen, V. Lemay, D. Weary, members of the vertebrate ecology research group at University of British Columbia (UBC), and comments from two anonymous reviewers.

### Funding

The project was supported by funding from the Natural Sciences and Engineering Research Council of Canada (NSERC), Environment and Climate Change Canada, including the Space for Habitat Project and Science Horizons Intern Program, Forest Renewal British

Columbia, and Tolko Limited to KM, and the Forest Investment Account Forest Science Program (Graduate Student Pilot Project and Mountain Pine Beetle Initiative Graduate Research Fund), the Southern Interior Bluebird Trail Society, a Junco Technologies Award (Society of Canadian Ornithologists and Bird Studies Canada), NSERC and University of British Columbia (Doctoral Fellowship, Pacific Century Graduate Scholarship) to ARN. The funders had no role in study design, data collection and analysis, decision to publish, or preparation of the manuscript.

## Grant Disclosures

The following grant information was disclosed by the authors:
Natural Sciences and Engineering Research Council of Canada.
Environment and Climate Change Canada.
Space for Habitat Project and Science Horizons Intern Program.
Forest Renewal British Columbia.
Tolko Limited.
Forest Investment Account Forest Science Program.
Mountain Pine Beetle Initiative Graduate Research Fund.
Southern Interior Bluebird Trail Society.
Junco Technologies Award.
NSERC Doctoral Fellowship.
University of British Columbia Pacific Graduate Scholarship.

## Competing Interests

The authors declare there are no competing interests.

## Author Contributions

- Andrea R. Norris conceived and designed the experiments, performed the experiments, analyzed the data, prepared figures and/or tables, authored or reviewed drafts of the article, and approved the final draft.
- Kathy Martin conceived and designed the experiments, performed the experiments, authored or reviewed drafts of the article, and approved the final draft.
- Kristina L. Cockle analyzed the data, prepared figures and/or tables, authored or reviewed drafts of the article, and approved the final draft.

## Animal Ethics

The following information was supplied relating to ethical approvals (*i.e.*, approving body and any reference numbers):

Approval for this research was provided by the Animal Care Committees of the University of British Columbia and Environment and Climate Change Canada, in accordance with the University of British Columbia and Environment and Climate Change Canada's Scientific and Animal Care Committees.

## Data Availability

The raw data and R code used for statistical analysis are available in the Supplementary Files.

## Supplemental Information

Supplemental information for this article can be found online at http://dx.doi.org/10.7717/peerj.14327#supplemental-information.

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
