# Peer review of "Weather and nest cavity characteristics influence fecundity in mountain chickadees"

_PeerJ, doi:10.7717/peerj.14327_

## Round 0.1 · original submission · Major Revisions

We received two consistent reports for the paper. Both include detailed comments and suggestions. Please address the issues raised by the reviewers and provide a detailed response letter.

Reviewer 1 ·

Basic reporting

Writing is good and generally clear

Experimental design

A lot of effort went into gathering data on food, surveying potential predators, measuring nest characteristics, getting environmental factors, and figuring out the appropriate model types to use. The samples sizes are not large, but are reasonable assuming the nests found (relatively few in some years) were random and representative of those produced in each year. The paper seems appropriate for the aims and scope of PeerJ.

Validity of the findings

Results (Tables 3 & 4). I like the approach of comparing models with sets of variables encompassing different levels of factors. What I’m unclear on is what was used as the response variable. Take the first set of analyses in Table 3, where “laying date” is the response variable. Like all the response variables, laying date is a property of each nest, but can also be averaged across all nests within a site or all nests in the region (within a particular year). So the question is: was each nest considered an independent datapoint in each of the models? In other words, for the first regional climate model, where “laying date ~ winter snow depth + spring temperature + winter storms”, was each nest considered an independent sample? Or was there only one line for each year, in which case “laying date” was (presumably) averaged across all nests within the region [or site] each year? The first of these seems weird, since the fixed effects would presumably have been the same for all nests within a year, but if the latter, then the models cannot be compared using dAICc values, since one can only do this when the response variables are identical (whereas “mean laying date across all nests in the region” is not the same as “laying date for each individual nest”). Please clarify or reanalyze.

Additional comments

The paper presents the results of a nice study of reproductive success in mountain chickadees nesting in natural cavities over a 12 year period. Most studies of secondary-cavity-nesting species are done in artificial cavities, making this one really quite unusual. A lot of effort went into gathering data on food, surveying potential predators, measuring nest characteristics, getting environmental factors, and figuring out the appropriate model types to use. The samples sizes are not large, but are good assuming the nests found (relatively few in some years) were random and representative of those produced in each year.

My only substantive comment is in regards of the analyses (under “Results, below”). Hopefully I have explained the issue clearly. The bottom line is that if the analyses were not done correctly, the dAICc comparisons will be bogus, the analyses need to be redone, and the paper needs to be rewritten appropriately. Specifically, if the response variable was not the same for all models within comparisons listed in Table 2, then some other method needs to be used to compare the relative importance of the different sets of variables at different scales than the dAICc values. Among other things, this could fairly easily explain the (apparently) generally poor showing of food, predators, and competitors as factors influencing most of the reproductive measures.

Specific comments:

Line 29. The fact that the study was done with birds nesting in natural cavities should be emphasized; such studies of secondary-nesting species are not common!

Lines 56-58. I’d delete the first sentence of the introduction; start with “annual fecundity” and save all the weather effects for where they start on line 61 (and continue through the end of the paragraph).

Line 80. I am a strong opponent of “but see”s. Either tell readers why they should see those papers or delete. Otherwise, it just looks like you’re contradicting yourself (which in a way you are).

Lines 128-130. I’m not sure that the amount of data gathered here qualifies as an exemplar of the “data-intensive” approach advocated by Kelling et al. (2009), who were (no doubt) trying to push eBird and the massive amounts of data it's gathering. The data used here is good, but hardly in the category of “big data” as it’s understood in this day and age.

Line 149. Shouldn’t this just be “ranged from 7 to 32 ha”?

Line 267. Need to explain “FLNR”.

Line 297. I appreciate the citation but I suspect that the relationship between conifer seed production and weather is more complex than what was suggested by Koenig & Knops (2000), who weren’t really focusing on environmental effects on masting. There’s lot of more recent literature on this that the authors might want to look into, depending on what the most abundant conifer species in the study site are.

Line 312. I believe the appropriate terminology of something along the lines of “models were considered to have support if their dAICc <2”.

Lines 317-318. What did you do if 2 variables were highly correlated? (How did you determine which to use?)

Line 326. “Bobyqa”? Please explain.

Line 336-338. I believe the sample size issue was previously mentioned.

Results (Tables 3 & 4). I like the approach of comparing models with sets of variables encompassing different levels of factors. What I’m unclear on is what was used as the response variable. Take the first set of analyses in Table 3, where “laying date” is the response variable. Like all 4 response variables, laying date is a property of each nest, but can also be averaged across all nests within a site or all nests in the region (within a particular year). So the question is: was each nest considered an independent datapoint in each of the models? In other words, for the first regional climate model, where “laying date ~ winter snow depth + spring temperature + winter storms”, was each nest considered an independent sample? Or was there only one line for each year, in which case “laying date” was (presumably) averaged across all nests within the region [or site] each year? The first of these seems weird, since the fixed effects would presumably have been the same for all nests within a year, but if the latter, then the models cannot be compared using dAICc values, since one can only do this when the response variables are identical (whereas “mean laying date across all nests in the region” is not the same as “laying date for each individual nest”). Please clarify or reanalyze.

Lines 432-433. Capitalize common names of birds (or not) consistently.

Line 508. I missed what the claim of “exceptional variation in annual fecundity” was based on. What were mountain chickadee compared to? My sense is that wide annual variation in fecundity is the norm, not the exception.

Line 514-525. I guess it’s a nice (and novel) idea to suggest the potential advances that “Indigenous epistemologies” might afford (although an example would be helpful), but I’m not sure I’d recommend ending the paper by apologizing for the “reductionist western scientific framework” used by this paper (and pretty much all of us for our entire careers). (However, having said that, the nod to the indigenous peoples in the acknowledgements on lines 544-548 is great.) Also, see the comment above (lines 128-130) regarding the “data-intensive” approach statement.

Line 538. What, exactly, is “co-produced knowledge”?

Reviewer 2 ·

Basic reporting

Please find enclosed my review for “Direct and indirect effects of weather on fecundity in a disturbance-resilient species (#73334)”

This is an intriguing study that connects weather and climate variation to variation in reproductive output across years. Most of the questions represent commonly asked, but very important questions in this field, specifically, how does weather influence timing of breeding and resulting reproductive success, especially through its influence on food abundance. The more we can uncover about how climate will influence breeding dynamics in different populations, different species, and different habitats, the better we’ll be able predict how species will respond to climate change. So this study offers the potential for an important contribution to the field. I also commend the authors for their work studying this species in natural cavities, something that very few other Parid researchers do. Overall I think the manuscript has good potential but it still needs a fair bit of work before publication.

Major points:

Insects and food resources: the authors raise an important point that forest insect abundance is likely to influence mountain chickadee breeding dynamics. The authors clearly put a lot of time into measuring abundance of two insect groups that previous records indicate are food sources for mountain chickadees: spruce budworm and mountain pine beetle. This is very useful in determining the effect of these two insect groups on breeding dynamics but the authors also state there are many other insect orders that mountain chickadees consume as well. Therefore, the extrapolation of budworm and beetle abundance to forest insect abundance goes too far. Without direct observations of what the chickadees consumed during the study, either through visual observations or through fecal DNA, it does not make sense to claim that budworm and beetle abundance represents total forest insect abundance or total prey availability. This does not take away from the importance of their findings, but many parts of the manuscript need to be clarified to reflect that budworm and beetle abundance are what was measured, not total forest insect abundance.

For example in Figure 1, the “Prey (insects)” box should be changed to “Budworm and beetle abundance” or something similar. Another example, in the abstract, in the background section it makes sense to say “forest insects” as a broad introduction, but then in the methods of the abstract “forest insect abundance” should be clarified.

Excluding snow depth: On line 288 the authors state that they excluded snow depth after 15 March because it correlated with spring temperature. This alone is not valid reasoning for excluding snow depth. Snow depth at the beginning of breeding could be a major factor influencing timing of breeding if it influences insect abundance. Even though it’s correlated, many researchers argue that you cannot separate out the effects of correlated climate variables (Freckleton 2011). At the very least, both snow depth and temperature should be included in the model, then one removed at a time, then you could compare the models using the ‘anova’ function in R to determine which explains more variation. If one is obviously better than the other, then you could use that model. But that reasoning and comparison needs to be explained and presented.

Nest vs regional level analyses: The site vs nest vs regional scale question is interesting but the main question of the paper is what are the factors that influence breeding variables. Therefore, after doing the scale comparison it would be helpful to create a final model that included any of the variables that were significant in the scale-level models. For example, for nest survival, the reader doesn’t get to see the effect of 2-yr temperature because only the nest-level model is reported in Table 4.

Simplifying the results: The results section is impressive but contains much information that is not necessary for the big picture and could be removed, or moved to supplemental tables. For example, the second part of the 354 sentence can easily be gathered from the figure and is not necessary.

Effect of conifer seed production: Like the effects of insect abundance, conclusions related to conifer seed production should be walked back to acknowledge that the authors did not actually measure seed or cone abundance. Relying on the Koenig paper is not a problem but it greatly reduces the conclusions that can be made about confer seeds themselves.

Figure 3 and Figure 5 – From what I understand, the authors ran the models on all years at the same time, with year as random effect. So it’s unclear to me the benefit of plotting the data by year. Doing so implies that the authors ran each year separately but this does not appear to be the case. These figures should re-constructed with all the years on one plot. Yearly figures might be useful for supplementary materials.

Line comments:

52 – what results justify calling this a climate-resilient species?

53 – Is there a word missing from this line? The end of this sentence doesn’t work.

56 – This first paragraph is almost a list of facts with little “flow”. What is the main point here? Why are these facts interesting?

71 – I’m not sure what “fulsome” means without looking it up. Consider using an adjective that is more commonly used in scientific writing.

72 – This paragraph really seems like it’s about secondary cavity nesters and not about non-migratory species as the first (topic) sentence indicates.

81 – “plasticity in annual fecundity” doesn’t make sense. Why would individuals want to be plastic in their reproductive output? Wouldn’t they want to produce as many offspring as possible? Especially for a relatively short-lived species like chickadees. Please clarify what you mean by this phrase or consider removing it. Maybe you’re referring to first egg date?

91 – Are the large entrances woodpecker cavities? I think this is implied but it takes a bit of re-reading to confirm. It might be helpful to clarify. It’s pretty clear that the small cavities are nuthatch cavities.

94-95 – It’s not clear to me why this point is important: “shift in nest-site use signalled a positive legacy effect on secondary cavity nesting populations”. Is this for chickadees? Or for other species? The preceding point in this sentence is very cool.

97 – This paragraph is quite long and jumps around a lot. It might be easier for your reader to take everything in if it were split up. Maybe a paragraph focusing on possible negative influences, like competition and predation, and a separate paragraph focusing on more positive influences, like food, would help? Either way it could benefit from streamlining the ideas.

106 – Statement needs a citation or needs to be softened if masting events have not been shown to influence laying date, etc.

113-118 – This sentence is really long and changes tense part-way through.

126 – I’m not sure why justifying an observational approach is required. Observational data are important too!

129 – Arguably, most scientific papers are data-intensive. That’s not meant to be snarky, the rest of the sentence is great and drives home the impressive scope of this project. “Data-intensive” seems like an unhelpful buzzword.
147 – At this point in the paper the importance of Williams Lake isn’t clear. Maybe add on to the end of the sentence: “where the weather station is located” or something similar
173 – It might be helpful to define double-brooding at the end of this paragraph. Otherwise it doesn’t get defined until the results.

184 – Do mountain chickadees sometimes excavate?

200 – Six minute point counts are likely to underestimate the true number of chickadees and squirrels. This should be noted. I think it’s still a valuable measure even so.

215 – “monogamous” implies they exhibit no extra-pair mating. In reality mountain chickadees are probably socially monogamous.

226-229 – Citation for this claim about foraging range?

294-295 – Both points in this sentence need citations.

297 – “rainfall” should probably be “precipitation” unless this paper only looked at rainfall. Also were the species that the Koenig paper examined the same as those at your field sites? This should be clarified whether they’re the same or related species.

317 – Site should also be included as a random effect in your models, or if there’s good reason for not using it as such this should be explained.

317 – Year should probably also be run as an interaction with the fixed effect of interest. For example, in Figure 3 it looks like you might have different relationships for different years. So it might be worth running your models with a year interaction and without a year interaction and then presenting both since they ask slightly different questions.

375 – Why is the relationship between clutch size and daily nest survival interesting? It’s not clear to me why this question is relevant or what the underlying biology is here. Please explain.

Table 2 can probably be supplemental material.

Table 4 probably needs more statistics – did your models also give a z or t value? Are those not usually reported with confidence intervals? I can be corrected on this if I’m wrong.

References:
Freckleton, Robert P. "Dealing with collinearity in behavioural and ecological data: model averaging and the problems of measurement error." Behavioral Ecology and Sociobiology 65.1 (2011): 91-101.

Experimental design

no comment

Validity of the findings

no comment

Additional comments

no comment

---

## Round 0.2 · Minor Revisions

The reviewers still have some lingering comments. Please revise the paper and provide a detailed response letter. Thanks.

Reviewer 1 ·

Basic reporting

Paper reads well and explores an interesting question.

Experimental design

Well done; although see my concern below.

Validity of the findings

See my "additional comments" below.

Additional comments

The paper is really nice. However! I still worry about the statistical models. As I understand it, the authors used each nest as independent data points in all the models, which is fine (and addresses my previous concern). However, I worry about the statement that, by including "year" as a random factor, "response variables (and the variation explained by intra-annual variation) are essentially averaged across all nests within each year". This may be true, but I still can't help but think that there is a problem with a model where each nest is considered separately, but all of the fixed explanatory variables are the same for each nest in each year, as would be the case for all of the models at the regional and site levels. Specifically, the concern is that the models where the explanatory variables are the same (within years) are NOT comparable to the set of models (at the "Nest" level) where this is not the case, even with "year" being included as a random variable. Since the models at the "Nest" level far outperform the other models in each of the comparisons where it occurs, this strikes me as an issue that needs to be checked.

I am not a statistician by any means, and I admit that my concern is mostly based on a hunch. (It just doesn't seem right.) Nonetheless, I urge the authors to check with a real statistician concerning this before calling it a wrap. After all, it's at least possible my concern is justified, in which case the analyses really will need to be redone and the results will potentially be (totally) changed--something the authors will want to discover now rather than later.

Reviewer 2 ·

Basic reporting

The authors have done an excellent job of revising this paper. It’s now much more readable and interpretable, thank you for taking the time to do that. It should make an important contribution to this literature.

I mostly have very minor comments listed out by line below. The only comment I have that will take slightly more work is I still find the explanation for why clutch size is a useful predictor of nest survival to be lacking. It’s fine to include in the paper, but your predicted relationship for these two variables should be introduced in the introduction (or at least reference Table 1) and discussed more fully in the discussion. For example, in the discussion I think it only comes up in the first paragraph. The relationship is clear, but why do you think that relationship occurs? I do appreciate the addition to Table 1 but slightly more context in the text would help as well.

Line 26 – Consider adding “and” before “fecundity”. Without a break the sentence gets a little bogged down with all the information.

130 - “so a single-species experimental approach was not ideal.” To me this sounds like you’re apologizing for not using an experimental approach. You could drop this clause and the paragraph would still read just fine.

243 – Do we know for sure that winter is when they accumulate resources to breed? If so, including a citation here would be helpful. If it’s not been specifically shown, you could just add “likely” to that claim since it’s a probable hypothesis.

305-309 – Are winter storms usually snow and spring storms usually rain? Neither is specified but it might be helpful to add a little context at the end of this paragraph.

335 – The delta-AIC < 2 idea needs slightly more context. I assume you’re saying if a model has a delta-AIC within 2 values of the top model it is plausible.

346 – should “random” be “non-random” since you have repeated measures? Might be simpler just to drop “random”. “Structure” alone should be enough.

356 – effects is misspelled

Figure 3 caption – specification needed: are points offset on both the x and y axis? Or just one axis?

Figure 3 - since you treat storms as categorical in the caption, ideally your storm color key would also be categorical rather than a gradient.

Figure 4 – I still think this would be better as one panel but if you think that would be too difficult to interpret you could retain the multi-panel figure and add some clarification to the caption, something similar to: “The model was run on data from all years, but the model predictions are presented separately for each year to improve interpretability.”

Figure 6 caption – add offset on x and y axis specification.

Experimental design

no comment

Validity of the findings

no comment

Additional comments

no comment

---

## Round 0.3 · accepted · Accept

All reviewers' comments have been addressed. The paper can be accepted.

Reviewer 1 ·

Basic reporting

It's good that the authors went to the trouble of checking out my concerns regarding model selection with a statistician, and I'm happy to defer to his/her judgement. I have no further concerns regarding the MS, which, as I said originally, presents the results of a nice study of reproductive success in mountain chickadees nesting in natural cavities over a 12 year period.

Experimental design

No comment

Validity of the findings

No comment

Additional comments

No additional comments

Reviewer 2 ·

Basic reporting

The authors have addressed all of my points and the manuscript looks ready for publication. I only found one minor thing to change:

The Figure 3 caption describes "black lines" but this must be from an older version. The current lines are not black.

Experimental design

No comment

Validity of the findings

No comment

Additional comments

No comment